# A Surface-Enhanced Raman Spectroscopic Sensor Pen

Zejiang Song [1], Zhijie Li [2], Weishen Zhan [2], Wanli Zhao [3], Hsiang-Chen Chui [1,*] and Rui Li [2,*]

1 School of Optoelectronic Engineering and Instrumentation Science, Dalian University of Technology, Dalian 116024, China; songzj972916@163.com

2 School of Physics, Dalian University of Technology, Dalian 116000, China; lajitx@yahoo.com (Z.L.); zhanwsh@dlut.edu.cn (W.Z.)

3 Science and Technology on Electro-Optical Information Security Control Laboratory, Tianjin 300308, China

* Correspondence: hcchui@dlut.edu.cn (H.-C.C.); rli@dlut.edu.cn (R.L.)

**Abstract:** Surface-enhanced Raman spectroscopy (SERS) is widely used as a detection method in scientific research fields. However, the method for creating SERS substrates often requires expensive equipment and involves a complex process. Additionally, preserving and effectively utilizing SERS substrates in the long term poses a challenging problem. In order to address these issues, we propose a new method for creating SERS substrates on various types of paper using a combination of a ballpoint pen and 3D printing. This method ensures a high enhancement factor and maximizes the utilization of the substrate. We achieved an enhancement factor of up to $8.2 \times 10^8$ for detecting R6G molecules, with a relative standard deviation of 11.13% for the Raman peak at 612 cm$^{-1}$ of R6G, demonstrating excellent SERS sensitivity and spectral reproducibility. Furthermore, we successfully detected thiram at a concentration as low as $10^{-8}$, which is lower than both the Chinese national standard and European standard.

**Keywords:** surface-enhancement Raman scattering; chemical sensors; nanoparticles; three-dimensional printing

## 1. Introduction

Surface-enhanced Raman scattering (SERS) has become one of the most attractive and powerful spectroscopic techniques for label-free ultra-sensitive detection of chemical and biological species. Each molecule has unique Raman spectral fingerprints. Refs. [1–3] SERS can effectively enhance their Raman signals and now have an important role in various fields, such as medical diagnostics, environment, and security applications. However, the qualities of SERS substrates determines the signal enhancement of SERS. Many SERS substrate preparation methods have been reported and are available, including electron beam lithography, chemical vapor deposition, nanoparticle (NP) deposition, and self-assembly NPs [4–7]. These exhibit excellent detection performances, but large-scale preparation always leads to laborious and quite expensive processes. Recently, paper-based SERS substrates have received a lot of attention. They have the advantages of being inexpensive; flow easily; are biodegradable, disposable, and lightweight; and are widely used [8–11]. And a great deal of research has been carried out with the aim of impregnating metallic NPs into paper using various processes, such as dip coating, inkjet printing, screen printing, and physical vapor deposition [10,12–14]. Hongk [15] and colleagues reported a SERS substrate. They applied an imidazolate molecular sieve skeleton layer onto gold NPs dipping paper, prepared using the dry plasma reduction method. 4-thioglytophenaldehyde was used as the probe molecule. The enhancement factor (EF) is $1.0 \times 10^6$. These processes require a more time-consuming immersion of the paper in concentrated NPs. Shuai [16] fixed gold nanostars onto filter paper to make a SERS substrate. And they used crystal violet as the Raman probe molecule. The EF is $1.2 \times 10^7$. However, this method requires the early processing of the filter paper. Processing steps are complex and time-consuming. Jun [17] fixed

Ag NPs onto a polyurethane (PU) sponge for a SERS substrate, using 4-aminothiophen as a Raman probe molecule. The EF was reported as $2.67 \times 10^6$. This flexible substrate improves the sample collection efficiency. However, the production process of the SERS substrate is more complicated. Pushkaraj [18] used a 2D SERS active silver nanowire network to solve the substrate storage problems by binding NPs to hydrogels. In this work [19], the low-cost and high-sensitivity-aligned SERS templates were prepared using 3D printing technology, thus demonstrating that 3D printing technology can be used to produce renewable SERS motifs and SERS arrays. Russo et al. [20] reported on a pen that can write on paper for flexible electronic devices. L. Polavarapu investigated a pen-on-paper (POP) method to fabricate SERS substrates. The SERS properties of plasma paper substrates prepared with three different types of NPs at three different excitation wavelengths were investigated [21].

In this work, we prepared silver (Ag) NP gel pens and successfully demonstrated them to be good SERS devices. Because Ag NPs have higher extinction cross-sections than gold NPs [22], they usually have a higher SERS efficiency. Therefore, we chose Ag NPs as the filling material of the silver glue pen. However, Ag NPs have strong chemical activity and are easily oxidized in the air. We improved the silver glue pen to effectively protect Ag NPs. Then, we studied several paper-based materials commonly used in the laboratory, tested a variety of paper platforms, and analyzed in detail the specific details of how this method affected Raman strength in the process of substrate preparation. SERS substrate was directly made with silver glue pen on paper-based substrate [23]. This method is convenient, rapid, and has a better enhancement effect than the traditional laboratory capillary sampling method.

In this work, R6G was chosen as the probe molecule for SERS analysis. The SERS immersion method was used to determine the amount of R6G, a highly efficient probe molecule commonly used for the determination of Raman spectral enhancement. The limit of detection (LoD) for R6G was 98.4 fmol/L, and the EF for a paper-based SERS substrate was calculated as $8.2 \times 10^8$ for R6G. Additionally, substrates were characterized via scanning electron microscopy (SEM). Formicide and methyl parathion are common pesticides in apple cultivation, which are harmful to the human body and can easily remain on the fruit's surface [20]. On this Raman detection platform, we detected the pesticide residues on fruit skins, which can be detected at a concentration of $10^{-8}$ mol/L of formicide pesticide residues. In the proposed method, the preserving Ag NPs gel becomes cheap, simple to produce, and has a good enhancement effect, which can be applied in the detection of low concentrations of sample detection.

## 2. Experiment

### 2.1. Preparation of Silver Nanoparticle Gel

We use the traditional method of reducing silver nitrate to prepare Ag NPs, because the Ag NPs prepared via this method have high purity, no obvious heterogeneous peaks in Raman spectroscopy, and the enhancement effect is excellent, and we optimized the experimental method, as shown in Figure 1. The specific reaction process is as follows: silver nitrate reacts with sodium citrate in the system, through the production of thermal hydration electrons and reduced free radicals, hydration electrons, and reducing free radical reduction solution. The silver of these elements serve as the crystal core. With the increase in the reaction time, the crystal nucleus grows slowly, and then the gradually increasing silver NPs form a stable micelle adsorption. The Ag NPs prepared in this experiment were in the form of spheres with high homogeneity under SEM and an average diameter of 52–54 nm, as shown in Figure 1b.

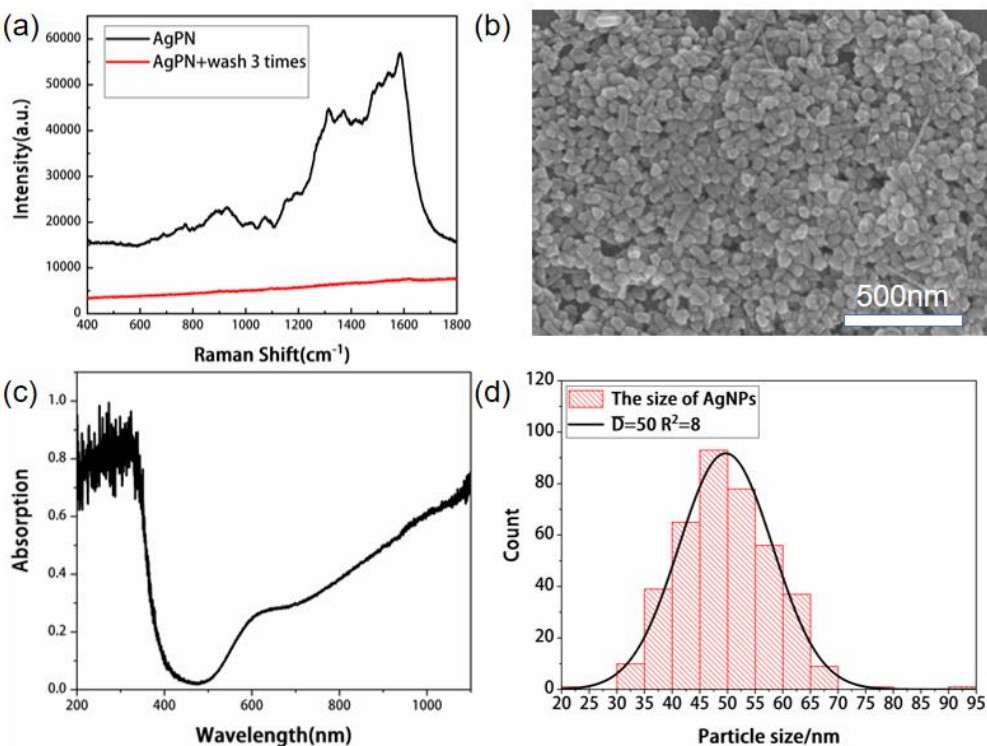

**Figure 1.** (**a**) Raman spectrum of Ag NPs before and after water washing. (**b**) SEM image of Ag gel by microwave method. (**c**) Absorption spectrum of Ag gel using the microwave method. (**d**) Statistical graph of particle size distribution of Ag gel.

In order to verify the sensitivity of the synthetic silver nanoparticle-coated paper-based SERS sensor, the approximate EF is estimated [24], and the calculation enhancement factor is an important index of the performance characterization of surface-enhanced Raman scatter. Most of the test methods involve placing the samples of the probe molecule into the SERS-active substrate glass or silicon and then drying it. The formula for calculating the enhancement factor is:

$$EF = \frac{I_{SERS}/C_{SERS}}{I_{RS}/C_{RS}} \tag{1}$$

where $C_{RS}$ and $C_{SERS}$ are the analyte concentrations of Raman scattering and analysis, respectively. $I_{RS}$ and $I_{SERS}$ are the signal intensity of Raman scattering and surface-enhanced Raman scattering corresponding to their concentrations, respectively.

### 2.2. Preparation of Ag NPs Gel Pen

The empty cartridge was cleaned using acetone, alcohol, and pure water in the ultrasonic cleaning device, and then air dried in a vacuum oven. The preparation of good glue Ag NPs was performed by injecting the gel into the empty cartridge, and the upper nano Ag gel injection height was about 0.5 cm of lithium base grease, as lithium base grease does not dissolve in water; so, both should be isolated from outside air to prevent the oxidation of the Ag gel through the oxygen in the air. The outflow rate of Ag NPs can also be controlled in order to avoid a liquid outflow rate that is too fast or too slow and will not pollute the Ag gel solution. The front end of the pen refill is equipped with a matching pen refill cap, which can completely isolate the air. When used, the Ag gel solution flows out evenly through the ball on the front end of the pen refill and is sealed with the pen refill cap after use. Finally, the Ag NPs gel pen was put into the refrigerator and stored for one month, after which SERS substrate was formulated again, and the preservation effect of the pen core on the Ag gel was detected.

### 2.3. Collection of the Raman Data

In this experiment, a Renishaw spectrometer was used to collect spectral data. The wavelengths were 532 nm, 633 nm, and 785 nm as incident excitation light. A 50-fold objective lens (MUL03501, Nikon, Japan) was used to focus the laser on the sample surface. The laser power was 75 mW, the integration time of each spectrum was 10 s, and the resolution of all spectra was 1 cm$^{-1}$. The experiment was conducted by increasing the diameter of the laser spot and reducing the laser power density, reaching the sample surface using the "defocus" option of the InVia Raman microscope (value: 100%). In this work, WIRE4.4 software was used to remove ghost lines and smooth spectral data. In order to filter out some noise, the Savitsky–Golay method of the software was used to smooth spectral data. The Savitzky–Golay filter was first proposed by Savitzky and Golay in 1964. It is widely used in data flow smoothing and denoising and is a filtering method based on the local polynomial least square fitting in the time domain. The biggest characteristic of this filter lies in that the shape and width of the signal can be ensured unchanged while the noise is removed. The number of window points is 15 and the polynomial order is 2. The fluorescence spectral data typically would be mixed with the SERS data. In order to prevent the background fluorescence signal spectrum, baseline correction must be conducted before for data statistics. In the baseline correction based on the asymmetric least-square smoothing baseline model, the asymmetry factor is set as 0.001, the threshold is set to 0.05, the smoothing factor was set to 4, and the number of iterations was set to 10. In the spectral experiment of this paper, for the R6G probe molecule, the Raman peak at 613 cm$^{-1}$ is mainly counted, because the intensity of this Raman peak can easily represent the intensity of the whole spectral data [25].

## 3. Results

### 3.1. Test of Silver Pen

The Ag NPs gel pen (Figure 2a) was stored at 4 °C for one month, and then the Raman spectrum test was conducted. The Raman spectra was shown in Figure 2b. The enhancement effect of the Ag gel pen on R6G was almost unchanged, indicating that the Ag gel pen had a good preservation effect. In order to select the laser wavelength with the best SERS effect, the spectra of R6G on air-laid paper at 532 nm, 633 nm, and 785 nm were detected, as shown in Figure 2c. The 532 nm laser wavelength had the strongest enhancement effect on the sample, followed by 633 nm, while the Raman peak of the sample was almost invisible at 785 nm. Therefore, the laser wavelength of 532 nm was used in this experiment. Meanwhile, as shown in Figure 2d, by comparing the SERS effect of undiluted high-concentration Ag gel and Ag gel diluted twice with deionized water on $10^{-7}$ mol/L R6G, we found that high-concentration Ag gel had a better enhancement effect, so high-concentration Ag NPs were used for the experiment.

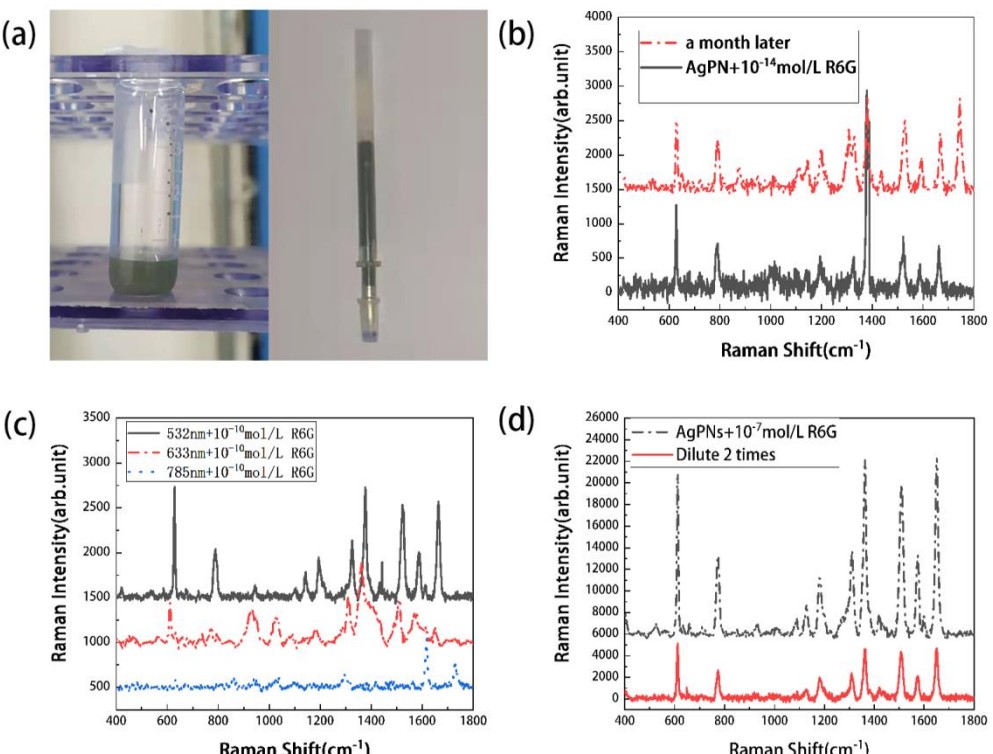

**Figure 2.** (**a**) Ag NPs and silver nano-pen. (**b**) SERS with fresh and a month preserved silver nano-pen. (**c**) SERS at different laser wavelengths. (**d**) SERS with different silver gel concentrations.

### 3.2. Selection of Paper Type

Optical images of different types of paper substrates are shown in Figure 3a–c. Both types of paper are composed of dense paper-based capillary fibers, with glass fibers exhibiting finer filaments. From the SEM images of Ag NPs on the papers, as shown in Figure 3d,e), the Ag NPs on the paper-based fiber was random distributed and fill in the gaps in paper materials. The laser beam can penetrate the paper materials and Ag NPs gel, and optical excitations are used to produce higher SERS EF [26]. For dust-free paper and glass fibers, we tested our own Raman peak using a silver glue pen. As shown in Figure 3f, it can be seen that no results from this paper have a sharp Raman peaks, indicating that the SERS substrate has no noise interference and is excellent substrate material. According to the comparison in Figure 3g, the SERS-EF of paper-based materials is significantly higher than those of the capillary tubes and slides commonly used in the laboratory. We can see that in the absence of the silver glue enhancement, the dust-free paper can detect 1 mol/L of R6G compared with the glass sheet, proving that the loose and porous paper-based material of the dust-free paper can help in SERS signal enhancement. The loose and porous structure of the paper-based material is conducive to the enhancement of the SERS signal. At the same time, the paper-based material does not interfere with the noise of the sample molecules. We believe that the paper-based material is an excellent material for making the SERS substrate.

The Raman spectra of different kinds of paper are shown in Figure 4. The enhancement effect from high to low is as follows: glass fiber > air-laid paper. However, the thickness of glass fiber is large, and the NPs are immersed between layers of glass fiber. After soaking the samples to be tested, the effect of Raman spectroscopy collection is not ideal, and the transmittance is very poor; thus, it is difficult to focus on the fiber and the spectrum collection is difficult. Air-laid paper has strong hydrophilicity and thick fiber, which is conducive to laser irradiation and Raman spectrum acquisition. Due to the unknown chemical substance in the base body of A4 paper, the Raman peak of the A4 paper itself will appear when it is detected as being below $10^{-8}$ mol/L, so the sample concentration of

the A4 paper below $10^{-8}$ mol/L cannot be detected. In conclusion, we chose the air-laid paper for the best SERS paper substrate. We compared three kinds of paper, and found that dust-free paper has loose porous structure. This structure is conducive to laser focusing on paper-based fiber, to laser irradiation and data acquisition, and to dust-free paper under the enhancement of silver glue; thus, we think that dust-free paper made from paper-based SERS substrate is excellent paper-based material.

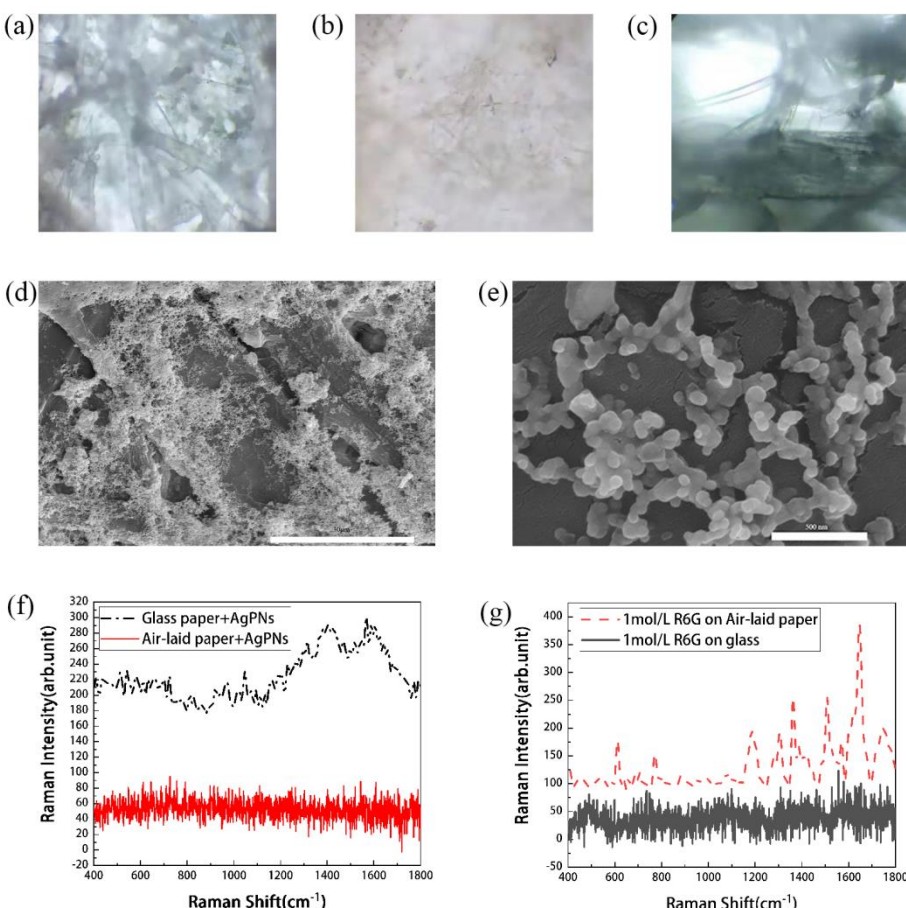

**Figure 3.** (**a**) A4 paper. (**b**) Glass fiber. (**c**) Air-laid paper. (**d**) SEM of A4 paper coated with silver gel (low magnification). (**e**) SEM of A4 paper coated with Ag gel (high magnification). (**f**) Raman spectra of two paper bases coated with Ag gel. (**g**) Comparison of the enhancement effect between capillary and air-laid paper.

By testing sensing performances, a stable and indicated absorption peak around 613 cm$^{-1}$ was found. The EF of the paper-based SERS sensor coated with Ag NPs was estimated using rhodamine 6G (R6G). The same volumes of $10^{-14}$ mol/L and 1 mol/L of R6G solutions in the presence and absence of Ag NPs were dropped onto glass plates, left to dry, and used for I$_{RS}$ and I$_{SERS}$ signal acquisition, respectively. The I$_{RS}$ and I$_{SERS}$ measurements were 74.5 and 139.4, respectively, based on the peak at R6G 613 cm$^{-1}$. The approximate EF estimate of $5.34 \times 10^{13}$ indicates that the resulting SERS sensor has high SERS efficiency.

We calculated the LOD of the detection limit of air-laid paper for R6G solution as follows:

$$\text{LOD} = \frac{3\sigma}{k} \tag{2}$$

The LOD is the detection limit, σ is the standard deviation of blank probe sample measurement, and k is the slope of calibration curve. For air-laid paper, σ = 4.784, k is 485.8, and LOD is 98.4 fmol/L, indicating that the SERS sensor has high SERS efficiency.

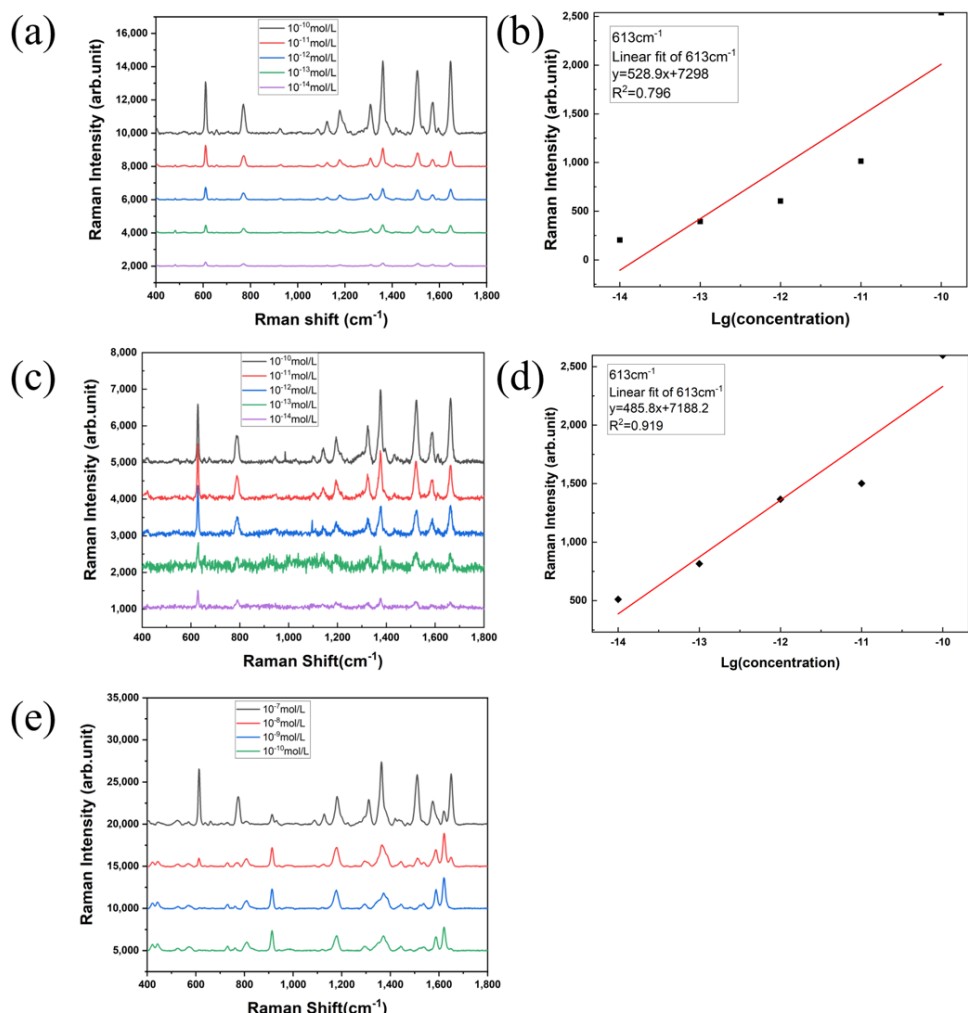

**Figure 4.** (**a**) Raman spectrum R6G detection limit of glass fiber. (**b**) Fitting curve of R6G concentration on glass fiber. (**c**) Raman spectrum R6G detection limit of air-laid paper. (**d**) Fitting curve of R6G concentration of air-laid paper. (**e**) R6G detection limit Raman spectrum of A4 paper.

The signal uniformity can be obtained with the relative standard deviation (RSD). The reproducibility test results of paper-based SERS signal are shown in Figure 5, and the RSD of Raman intensity at 613 cm$^{-1}$ was 11.13%, which was better than the effect seen in some published articles [27]. We can draw the conclusion that the particle size distribution on the surface of the air-laid Ag NP paper is evener than that of the air-laid paper substrate with good reproducibility. Therefore, as the paper test method in this paper is very simple and fast, and the paper base material is very simple and cheap to use, disposable SERS base material is used for the paper base SERS. So, we do not need to test its stability. Therefore, the stability of paper SERS sensors depends on the stability of Ag NPs. The above paper has demonstrated that silver glue pen can store Ag NPs well for more than one month, so the stability of the paper-based SERS sensor discussed in this work is more than one month.

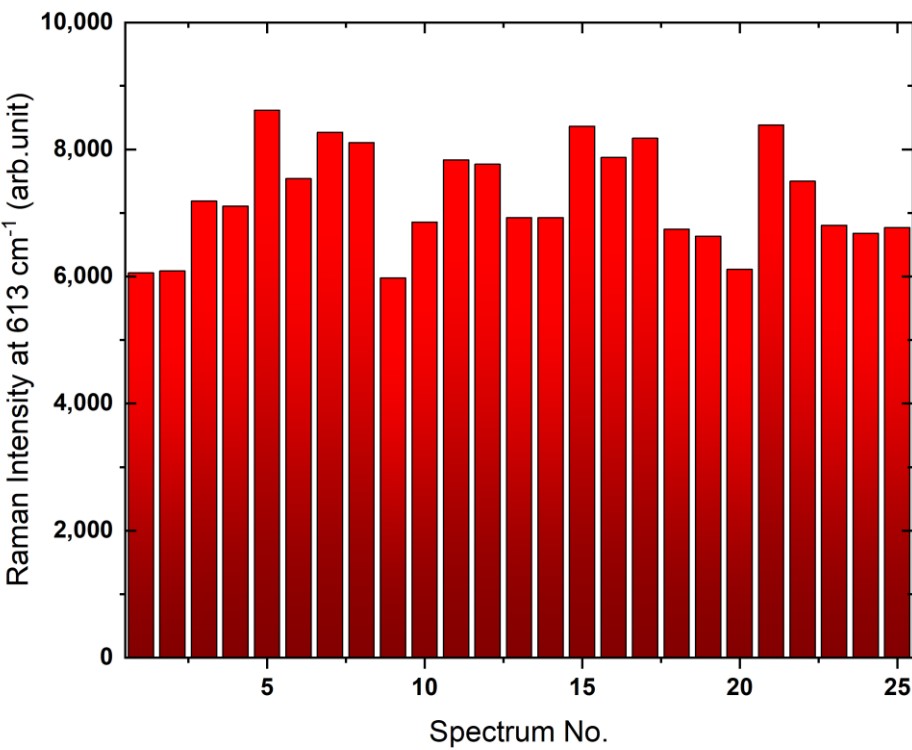

**Figure 5.** The Raman spectral homogeneities of the R6G.

### 3.3. Application Examples: 3D-Printed SERS Array

In order to test and practice the practical application value of the detection platform based on paper base and Ag gel pen, we carried out the pesticide detection experiment and SERS platform experiment using 3D printing, and realized the detection of $10^{-8}$ mol/L thiram pesticide and SERS arrays.

Figure 6a is the photo of 3D printer and SERS platform. The Ag NPs gel pen and 3D printer were combined to create the SERS array. The handwriting width is about 0.3 mm, the pattern can be designed using the printer's code program, and thus the mapping test was carried out. Figure 6b shows the 5 × 5 SERS array printed on 16 cm$^2$ A4 paper. Raman detection can be performed on the samples by dropping the sample solution onto the silver gel pattern points. Therefore, this design can test a large number of samples at one time. WIRE4.4 software was used to map and scan the cross-stars. Step sizes were 100 microns in both x and y directions. The Raman spectra of R6G at every point on the whole cross-star were scanned in this research. In the process of data processing, the data intensity of the Raman peak at 613 cm$^{-1}$ in the Raman spectral data of each point is selected to represent the intensity of the whole Raman spectrum. Figure 6c is the optical image of a single point in the array, represented by the Raman signal intensity at 613 cm$^{-1}$. The mapping diagram is shown in Figure 6d. The signal in the pattern area where the Ag gel is located is bright and red. The precision of the patterns drawn by this method can reach sub-millimeter accuracy (0.3 mm), and the patterns can be recognized in the mapping diagram. Therefore, it can be considered that high-precision SERS array mapping can be performed using 3D printers. Based on this, I can conclude that printing other SERS patterns is completely feasible. In conclusion, this paper enables SERS templates to print with low cost, high sensitivity, neat arrangements, and high accuracy using a 3D printer. The volume, concentration, and water content of the NPs at the SERS point on the high-precision SERS template are the same, which is conducive to the standardization of SERS substrate and the reproducibility of the SERS substrate.

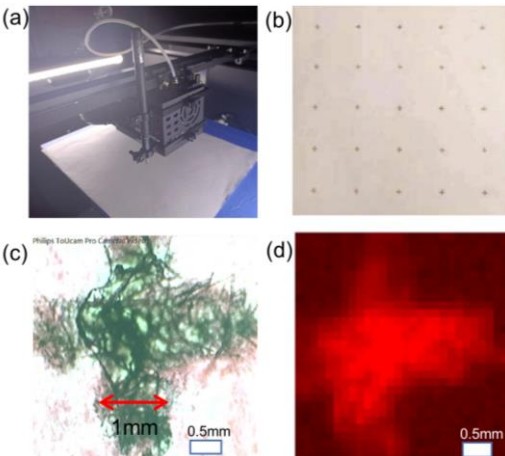

**Figure 6.** (**a**) Photo of 3D printer and SERS platform. (**b**) The 5 × 5 SERS array created with a Ag gel pen and 3D printer. (**c**) Single-amplified optical graph in the 5 × 5 SERS array. (**d**) Exact mapping image.

### 3.4. Application Examples: Pesticide Detection

The purchased 80% thiram were prepared into thiram solution with effective concentrations of $1 \times 10^{-3}$, $10^{-4}$, $10^{-5}$, $10^{-6}$, $10^{-7}$, and $10^{-8}$ mol/L, respectively. We applied the pesticide solution of standard concentration on the peel. After the pesticide was dried, we wiped the position of pesticide residue with the weight paper treated with the pen, and then conducted the Raman spectrum test. The minimum detected pesticide residue of formicide was $10^{-8}$ mol/L (as shown in Figure 7a), which is lower than the China national standard (GB 2763-2012) [28] of $2.435 \times 10^{-6}$ mol/L and the European standard of $9.74 \times 10^{-7}$ mol/L (European Community, Commission Directive 2007/57/EC) [29]. So, the detection method is effective and feasible, and the time from sample preparation to data processing can occur within 30 min, is efficient and simple, and can be used for on-site real-time detection. Figure 7b shows a linear relationship between the log value of formic double concentration and the intensity of Raman spectrum.

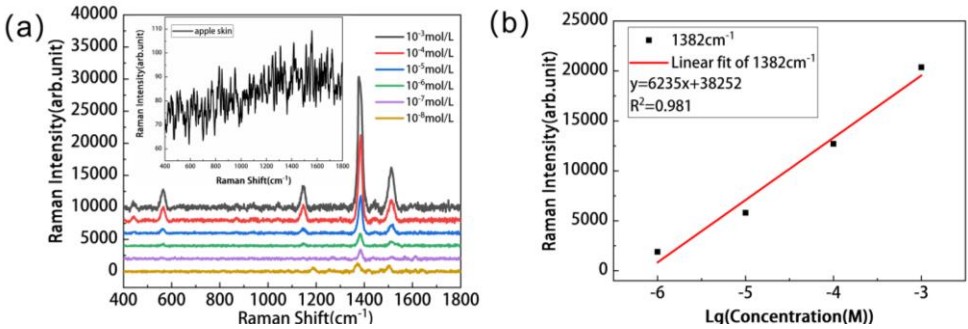

**Figure 7.** (**a**) Raman spectra of different concentrations of formicide on apple peel. (**b**) Fitting curve of formic double concentration on apple peel. The horizontal coordinate is the log of concentration.

### 4. Conclusions

In this work, the storage and use of Ag NPs gel devices was introduced, with different kinds of paper base platforms. Two practical examples were proposed: pesticide residue detection and the 3D-printed SERS array. Made with the method of the Ag NPs gel pen, Ag NPs are preserved from the air, then placed in an environment of low temperature (cold storage), thus greatly improving the stability of silver sol. Experimental results show that they can last a month under these conditions; therefore, this Ag gel made from a Raman-enhanced base is very convenient, requiring only ink written on a paper base. And compared the different kinds of paper bases, two effective practical application scenarios,

pesticide residue detection and a 3D-printing SERS array, were also discussed. The pesticide residues of Formicide on apple skin was detected, which can be as low as $10^{-8}$ mol/L, which is far lower than the Chinese national standard and the European standard. The SERS array made with a 3D printer can solve the problem of the simultaneous testing of a large number of samples. The thiram detections on the agricultural products are on-going.

**Author Contributions:** Conceptualization, Z.S. and Z.L.; methodology, W.Z. (Wanli Zhao); software, W.Z. (Wanli Zhao); validation, Z.S., Z.L. and W.Z. (Weishen Zhan); data curation, R.L.; writing—original draft preparation, R.L.; writing—review and editing, H.-C.C. All authors have read and agreed to the published version of the manuscript.

**Funding:** The Fundamental Research Funds by the Fundamental Research Funds for the Central Universities (DUT22LAB606 and DUT23YG205).

**Data Availability Statement:** The data presented in this study are available in the article.

**Conflicts of Interest:** The authors declare no conflict of interest.

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
