# Peer review of "A Surface-Enhanced Raman Spectroscopic Sensor Pen"

_inventions, doi:10.3390/inventions8060156_

Round 1

Reviewer 1 Report

Comments and Suggestions for Authors

The authors describe a new class of gel pen-based SERS nanosensors. The synthesis strategy is interesting since it allows the protection of Ag nanoparticles from oxidation and aging. The detection of the formicide pesticide on apple skin and a comparison with the Chinese standards has been achieved by SERS. Thus, the results seem to be quite promising, but the claims of the authors are not completely supported by the experimental results. Before acceptance, the following problems should be addressed.

1) The authors showed best results with the paper-based platforms. The authors failed to discuss the reason behind this enhancement using this type of sensor. One can see in Figure 3 aggregates of Ag nanoparticles, which are closely separated. Therefore, we can note a gap effect that might enhance the SERS signal. On the other hand, on other places on the paper, the number of nanoparticles is low. So, it is controversial if we compare to the sample shown in figure 1 where the gaps are quite smaller.

2) Low concentrations were detected and an enhancement factor of 1014 was determined. The calculation method considering the number of molecules detected and the excited surface, etc. should be described with more details.  

3) In addition, reproducibility of the SERS results should also be described. I must emphasize that reproducibility in the production of sensors is one of the major problems in the application of the technique.

5) According to figure 1 & 3, Ag nanoparticles are polydisperse in shape and size, so that the SERS response might change from zone to another on the platforms. How do the authors explain the signal reproducibility in figure 5.

6) The intro is too general; the authors failed to clearly state the research question, the key findings reported and the novelty aspects of this work.

7) There is an error in page 7, line 211. The sentence is not clear.

For all these points, I found the paper suitable for publication in Inventions journal after major revisions.

Reviewer 2 Report

Comments and Suggestions for Authors

Please see attached review comments.

Comments on the Quality of English Language

The paper contains numerous grammatical errors that significantly impact its readability and accessibility. To enhance the quality of the paper, I strongly recommend a thorough proofreading and revision of the text prior to resubmission. Additionally, many sentences should be refined to improve overall clarity and coherence.

Reviewer 3 Report

Comments and Suggestions for Authors

The paper proposes a new method for creating SERS substrates on various types of substrates using a combination of a ballpoint pen and 3D printing. The paper is well organized and partially wel written. However, I found  some shortcomings and inaccuracies in the results.

A EF of 13 orders of magnitude are not justified by the results, nor by any theroretica-numerical support. Realistically the authors obtain an EF of 10^6 10^7, that is anyway a good result.

The roughness of paper substrates is a critical quantity that the authors must characterize carefully. This parameter affects the SERS response, so that the authors must justify the SERS response as a function of paper roughness, before and after the AgNP deposition.

Minor changes regard

1) the modification of greater visibility in the text to the result on thiram, even if the work seems more oriented towards methodology than application, this result must be made more visible,

2) Savitzy-Golay is ofter written in uncorrect way,

3) "LOD for R6G was 0.00984 pmol/L" on page 2 must be justified, in addition "EF for a paper-based SERS substrate was 5.341013 for R6G" is not clear.

Comments on the Quality of English Language

The English is very readable and clear, some typos here and there, but they do not change the understanding and meaning of the results and methodology described. However, since I am not a native speaker, I am not qualified enough to express a definitive opinion on the quality of the English used, which still seems good to me.

Round 2

Reviewer 1 Report

Comments and Suggestions for Authors

After reviewing the second version of the manuscript and authors responses. 

I recommend the article publication in inventions journal in the present form.

Reviewer 2 Report

Comments and Suggestions for Authors

The authors have addressed my comments adequately. I recommend this paper for publication. 

Reviewer 3 Report

Comments and Suggestions for Authors

Tne authors have addressed all the issues evidenced after first revision.